# Smoothed Differential Privacy

**Ao Liu**                                                                                                          *aoliu.cs@gmail.com*
*Core Machine Learning, Google*

**Yu-Xiang Wang**                                                                                           *yuxiangw@cs.ucsb.edu*
*Department of Computer Science, UC Santa Barbara*

**Lirong Xia**                                                                                                      *xialirong@gmail.com*
*Computer Science Department, Rensselaer Polytechnic Institute*

**Reviewed on OpenReview:** *https://openreview.net/forum?id=CviCLt44Em*

## Abstract

Differential privacy (DP) is a widely-accepted and widely-applied notion of privacy based on worst-case analysis. Often, DP classifies most mechanisms without additive noise as non-private (Dwork et al., 2014). Thus, additive noises are added to improve privacy (to achieve DP). However, in many real-world applications, adding additive noise is undesirable (Bagdasaryan et al., 2019) and sometimes prohibited (Liu et al., 2020).

In this paper, we propose a natural extension of DP following the worst average-case idea behind the celebrated smoothed analysis (Spielman & Teng, 2004). Our notion, *smoothed DP*, can effectively measure the privacy leakage of mechanisms without additive noises under realistic settings. We prove that any discrete mechanism with sampling procedures is more private than what DP predicts, while many continuous mechanisms with sampling procedures are still non-private under smoothed DP. In addition, we prove several desirable properties of smoothed DP, including composition, robustness to post-processing, and distribution reduction. Based on those properties, we propose an efficient algorithm to calculate the privacy parameters for smoothed DP. Experimentally, we verify that, according to smoothed DP, the discrete sampling mechanisms are private in real-world elections, and some discrete neural networks can be private without adding any additive noise. We believe that these results contribute to the theoretical foundation of realistic privacy measures beyond worst-case analysis.

## 1 Introduction

*Differential privacy (DP)*, a *de facto* measure of privacy in academia and industry, is often achieved by adding additive noises (*e.g.,* Gaussian noise, Laplacian noise, and the discrete noise in exponential mechanism) to published information (Dwork et al., 2014). However, additive noises are procedurally or practically unacceptable in many real-world applications. For example, presidential elections often require a deterministic rule to be used (Liu et al., 2020). In such cases, though, *sampling noise* often exists, as shown in the following example.

**Example 1** (**Election with sampling noise**). *Due to COVID-19, many voters in the 2020 US presidential election chose to submit their votes by mail. Unfortunately, it was estimated that the US postal service might have lost up to 300,000 mail-in ballots (0.2% of all votes) (Bogage & Ingraham, 2020). For the purpose of illustration, suppose these votes are distributed uniformly at random, and the histogram of votes is announced after the election day.*

A critical public concern about elections is: should publishing the histogram of votes be viewed as a significant threat to privacy? Notice that with sampling noise such as in Example 1, the (sampling) histogram mechanism can be viewed as a randomized mechanism, formally called *sampling-histogram* in this paper. The same question can be asked about publishing the winner under a deterministic voting rule with sampling noise.

The standard notion of DP (Definition 1 in Dwork et al., 2006a) measures the worst-case privacy leakage (see Section 2 for its formal definition and detailed discussions). At a high level, DP considers the worst-case input and worst-case output of the (random) mechanisms. $\epsilon$ and $\delta$ are DP's privacy parameters to measure the privacy leakage in the worst-case described above. At a high level again, $\epsilon$ is a mechanism-designer-decided threshold for the "acceptable amount of privacy leakage", while $\delta$ measures the probability that the threshold got exceeded. Thus, smaller $\epsilon, \delta$ represents stronger privacy guarantees. The usual requirements upon a private mechanism are $\epsilon = O(1)$ and $\delta = o(1/n)$. The requirement on $\delta$ is more strict than $\epsilon$ because $\delta$ measures the "failure probability".

If we apply DP to answer this question, we would then conclude that publishing the histogram *can* poses a significant threat to privacy, as the privacy parameter $\delta \approx 1$ (See Section 2 for the formal definition) in the following worst-case scenario: all except one vote are for the Republican candidate, and there is one vote for the Democratic candidate. Notice that $\delta \approx 1$ is much worse than the threshold for private mechanisms, $\delta = o(1/n)$, where $n$ is the number of agents (voters). Moreover, using the adversary's utility as the measure of privacy loss (see Section 2 for the formal definition), in this (worst) case, the privacy loss is large ($\approx 1$, see Section 2 for the formal definition of utility), which means the adversary can make accurate predictions about every agent's preferences.

However, DP does not tell us whether publishing the histogram poses a significant threat to privacy *in general*. In particular, the worst-case scenario described in the previous paragraph never happened even approximately in the modern history of US presidential elections. In fact, no candidates get more than 70% of the votes since 1920 (Leip, 2023), when the progressive party dissolved.

It turns out that the privacy loss may not be as high as measured by DP, where the privacy loss is measured by the adversary's utility (*i.e.*, how accurately the adversary can infer/predict the votes, see Justification 3 in Section 2 for its formal definition). To see this, we assume $0.2\%$ of the votes were randomly lost in the presidential elections of each year since 1920 (in light of Example 1). Figure 1 presents the adversary's utility under this assumption. It can be seen that the adversary's utility is very limited (at the order of $10^{-32}$ to $10^{-8}$, always smaller than the threshold of private mechanisms, $1/n$). In other words, the adversary cannot get much information from the published histogram of votes. Figure 1 also plots the database-dependent privacy parameter $\delta(x)$ (Definition 2, a DP-like $\delta$ parameter for a specific database). One can see that $\delta(x)$ is closely related to the adversary's utility and is also at the order of $10^{-32}$ to $10^{-8}$. Besides, we observe an interesting decreasing trend in the adversary's utility, which implies that the elections become more private in more recent years. This is primarily due to the growth of the voting population, which is exponentially related to the adversary's utility (Theorem 3). In Appendix A.2, we further show that the elections are still private when only $0.01\%$ of votes got lost.

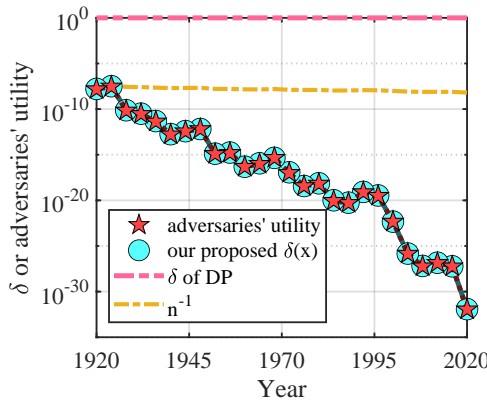

Figure 1: The privacy loss and adversaries' utility in US presidential elections. The smaller $\delta(x)$ is, the more private the election is. The smaller adversaries' utility is, the more private the election is.

As another example, for *neural networks* (NNs), even adding slight noise can lead to dramatic decreases in the prediction accuracy, especially when predicting underrepresented classes (Bagdasaryan et al., 2019). Sampling noises also widely exist in machine learning, for example, in the standard practice of cross-validation as well as in training (e.g., batch-sampling when training NNs).

Note that in all the above examples, the sampling noise is an "intrinsic" part of the mechanism. In comparison, the only purpose of adding additive noises is to improve privacy (with the cost of reducing accuracy) in most scenarios. As shown in these examples, the worst-case privacy according to DP might be too loose to serve as a practical measure for evaluating and comparing mechanisms with sampling noise (while without additive noise) in real-world applications. This motivates us to ask the following question.

> *How can we measure privacy for mechanisms with sampling noise under realistic models?*

The choice of model is critical and highly challenging. A model based on worst-case analysis (such as in DP) provides upper bounds on privacy loss, but as we have seen in Figure 1, in some situations, such upper bounds are too loose

to be informative in practice. This is similar to the runtime analysis of an algorithm—an algorithm with exponential worst-case runtime, such as the simplex algorithm, can be faster than some algorithms with polynomial runtime in practice. Average-case analysis is a natural choice of the model, but since "*all models are wrong*" (Box, 1979), any privacy measure designed for a certain distribution over data may not work well for other distributions. Moreover, ideally, the new measure should satisfy the desirable properties that played a central role behind the success of DP, including *composition* and *robustness to post-processing*. These properties make it easier for the mechanism designers to figure out the privacy level of mechanisms. Unfortunately, we are not aware of a measure based on average-case analysis that has these properties.

We believe that the *smoothed analysis* (Spielman, 2005) provides a promising framework for addressing this question. Smoothed analysis is an extension and combination of worst-case and average-case analyses that inherit the advantages of both. It measures the expected performance of algorithms under slight random perturbations of worst-case inputs. Compared with the average-case analysis, the assumptions of the smoothed analysis are much more natural. Compared with the worst-case analysis, the smoothed analysis can better describe the real-world performance of algorithms. For example, it successfully explained why the simplex algorithm is faster than some polynomial algorithms in practice (Spielman & Teng, 2004).

**Our Contributions.** The main merit of this paper is a new notion of privacy for mechanisms with sampling noise (and without additive noises), called *smoothed differential privacy* (*smoothed DP* for short), which applies smoothed analysis to the privacy parameter $\delta(x)$ (Definition 2) as a function of the database $x$. In our model, the "ground truth" distribution of agents is from a set of distributions $\Pi$ over data points, on top of which the nature adds random noises. Formally, the "smoothed" $\delta(x)$ is defined as

$$\delta_{\text{SDP}} \triangleq \max_{\vec{\pi}} \left( \mathbb{E}_{x \sim \vec{\pi}} \left[ \delta(x) \right] \right),$$

where $x \sim \vec{\pi} = (\pi_1, \cdots, \pi_n) \in \Pi^n$ means that for every $1 \leq i \leq n$, the $i$-th entry in the database independently follows the distribution $\pi_i$. We note that $\Pi$ is a parameter for smoothed analysis, not for the mechanisms $\mathcal{M}$. Table 1 compares the $\epsilon$ and $\delta$ parameters of smoothed DP and DP. The sampling histogram algorithm refers to Algorithm 2 (in Section 5.1), which samples $T = \lceil \eta \cdot n \rceil$ data without replacement and outputs the histogram of the sampled data. Appendix A.3 shows the detailed settings of Table 1. We present a high-level comparison between smoothed DP and other DP(-like) notions in Table 2. A more detailed comparison between DP, smoothed DP, and distributional DP can be found in Table 3 of Section 3.2.

| Notions | measures | without additive noise | with additive noise (of level $1/\epsilon$) |
|---|---|---|---|
| **Smoothed DP** | $\left(\epsilon, \delta_{\text{SDP}}\right)$ | $\left(\epsilon, \exp\left(-\Theta(n)\right)\right)$ | – |
| **(this paper)** | main message | **private** (under smoothed analysis) | private |
| DP (Definition 1, | $(\epsilon, \delta)$ | $(0, \eta)$ | $(\eta \cdot \epsilon, 0)$ |
| Dwork et al., 2006a) | main message | **non-private** (under worst-case analysis) | private (if $\epsilon$ is small) |

Table 1: Compare DP and smoothed DP for our motivation example (US presidential election) with a constant sampling rate $\eta$ (*e.g.*, $\eta = 1 - 0.2\%$ in the example). The $\epsilon$ part in $\delta_{\text{SDP}}$ is omitted for simplicity (see Theorem 3 for detailed discussions). The $(\epsilon, \delta_{\text{SDP}})$ for smoothed DP with additive noise is not shown in this table because it is already private without additive noise.

| Notions | database $x$ | output $\mathcal{S}$ | adjacent database $x'$ |
|---|---|---|---|
| **Smoothed DP** **(Definition 3, this paper)** | **smoothed analysis** | **worst-case analysis** | **worst-case analysis** |
| DP (Definition 1, Dwork et al., 2006a) | worst-case analysis | | |
| Gaussian DP or $f$-DP (Dong et al., 2019) | | | |
| Rényi DP Mironov (2017) | | average-case analysis (weighted by Rényi divergence or sub-Gaussian divergence) | |
| Concentrated DP (Bun & Steinke, 2016; Dwork & Rothblum, 2016) | | | |
| Bayesian DP (Triastcyn & Faltings, 2020) | | worst-case analysis | less informative adversary |
| Random DP (Hall et al., 2011) | less informative adversary | | |
| Distributional DP (Bassily et al., 2013) | | | worst-case analysis |

Table 2: The comparison between smoothed DP and other privacy notions.

**Theoretically,** we prove that smoothed DP satisfies many desirable properties, including two properties also satisfied by the standard DP: *robustness to post-processing* (Proposition 4) and *composition* (Proposition 5). Besides, we prove two additional properties for smoothed DP, called *pre-processing* (Proposition 6) and *distribution reduction* (Proposition 7). Based on pre-processing and distribution reduction, we propose an efficient algorithm (Algorithm 1) to calculate the privacy parameters for smoothed DP. We further show that, under smoothed DP, many discrete mechanisms with small sampling noise (and without any other noise) are significantly more private than those guaranteed by DP. For example, the sampling-histogram mechanism in Example 1 has an exponentially small $\delta_{\text{SDP}}$ (Theorem 3), which implies that the mechanism protects voters' privacy in elections—and this is in accordance with the observation on US election data in Figure 1. We also note that the sampling-histogram mechanism is widely used in machine learning (e.g., the SGD in quantized NNs). In comparison, smoothed DP implies a similar privacy level as the standard DP in many continuous mechanisms. We proved that smoothed DP and the standard DP have the same privacy level for the widely-used sampling-average mechanism when the inputs are continuous (Theorem 4).

**Experimentally,** we numerically evaluate the privacy level of the sampling-histogram mechanism using US presidential election data. Simulation results show an exponentially small $\delta_{\text{SDP}}$, which is in accordance with our Theorem 3. Our second experiment shows that a one-step *stochastic gradient descent* (SGD) in quantized NNs (Banner et al., 2018; Hubara et al., 2017) also has an exponentially small $\delta_{\text{SDP}}$. This result implies that SGD with gradient quantization can already be private in practice without adding any extra (additive) noise. In comparison, the standard DP notion always requires extra (additive) noise to make the network private at the cost of a significant reduction in accuracy.

**Related Work and Discussions.** There is a large body of literature on the theory and practice of DP and its extensions. We believe that the smoothed DP introduced in this paper is novel. To the best of our knowledge, none of the literature has proposed any DP-like notions for the mechanism without additive noises. A notable exception is distributional DP (Bassily et al., 2013), which considers a less informative adversary to provide a privacy measure for deterministic mechanisms. However, since distribution DP does not require any randomness in the mechanism, its privacy guarantee is much weaker than other DP-like notions. Rényi DP (Mironov, 2017), Gaussian DP (Dong et al., 2019) and Concentrated DP (Bun & Steinke, 2016; Dwork & Rothblum, 2016) target to provide tighter privacy bounds for the adaptive mechanisms. Those three notions generalized the $(\epsilon, \delta)$ measure of distance between distributions to other divergence measures. Bayesian DP (Triastcyn & Faltings, 2020) tries to provide an "affordable" measure of privacy that requires less additive noise than DP. With similar objectives, Bun and Steinke (Bun & Steinke, 2019) add noises according to the average sensitivity instead of the worst-case sensitivity required by DP. However, additive noises are required in (Bun & Steinke, 2019) and (Triastcyn & Faltings, 2020). Random DP (Hall et al., 2011) combines the high-level idea of distributional DP and Bayesian DP, which considers randomness in both the database $x$ and its neighboring database $x'$.

Appendix A.4 discusses the related works in the field of smoothed analysis, quantized neural networks, etc.

## 2   Differential Privacy and Its Interpretations

In this paper, we use $n$ to denote the number of records (entries) in a database $x \in \mathcal{X}^n$, where $\mathcal{X}$ denotes all possible values for a single entry. $n$ also represents the number of agents when one agent (one individual) can only contribute one record. We say that two databases $x, x'$ are *neighboring* (denoted as $x \simeq x'$) if one database can be gotten by replacing no more than one record from the other database. In the motivating example (Example 1), $\mathcal{X} = \{0, 1\}$, where $0$ represents a vote to one candidate while $1$ represents the vote to the other candidate. Database $x$ represents all voters' votes (a $n$-dimensional binary vector). A record represents a single dimension (*e.g.*, the $i$-th dimension) of $x$. Two databases are considered to be neighboring if no more than one voter's vote is different.

**Definition 1** (**Differential privacy**). *Let $\mathcal{M}$ denote a randomized algorithm and $\mathcal{S}$ be a subset of the image space of $\mathcal{M}$. Throughout this paper, image space" represents the space for the image of the (randomized) mechanism. $\mathcal{M}$ is said to be $(\epsilon, \delta)$-differentially private for some $\epsilon > 0, \delta > 0$, if for any $\mathcal{S}$ and any pair of neighboring database $x, x'$,*

$$\Pr[\mathcal{M}(x) \in \mathcal{S}] \leq e^\epsilon \Pr[\mathcal{M}(x') \in \mathcal{S}] + \delta, \tag{1}$$

*Notice that the randomness comes from the mechanism $\mathcal{M}$.*

DP guarantees immunity to many kinds of attacks (*e.g., linkage attacks* (Nguyen et al., Sep. 2013) and *reconstruction attacks* (Dwork et al., 2014)). Take *reconstruction attacks* for example, the adversary has access to a subset of the database (such information may come from public databases, social media, etc.). In an extreme situation, an adversary

knows all but one agent's records. To protect the data of every single agent, DP uses $\delta = o(1/n)$ as a common requirement of private mechanisms (Dwork et al., 2014, p. 18). To meet this requirement, a private mechanism (under DP notion) usually[1] need to include additive noises (*e.g.*, Gaussian noise, Laplacian noise, and the noise, which usually is discrete, in exponential mechanisms). Next, we recall two common interpretations/justifications on how DP helps protect privacy even in the extreme situation of reconstruction attacks. After that, we formally introduce the adversaries' utility in Figure 1 and use it to justify DP.

**Justification 1: DP prevents membership inference (Wasserman & Zhou, Mar. 2010; Kairouz et al., 2015).**
Assume that the adversary knows all entries except the $i$-th. Let $x_{-i}$ denote the database $x$ with its $i$-th entry removed. With the information provided by the output $\mathcal{M}(x)$, the adversary can infer the missing entry by testing the following two hypotheses:

$$\mathcal{H}_0\text{: The missing entry is } X \text{ (or equivalently, the database is } x = x_{-i} \cup \{X\}).$$
$$\mathcal{H}_1\text{: The missing entry is } X' \text{ (or equivalently, the database is } x' = x_{-i} \cup \{X'\}).$$

Suppose that after observing the output of $\mathcal{M}$, the adversary uses a rejection region rule for hypothesis testing[2], where $\mathcal{H}_0$ is rejected if and only if the output is in the rejection region $\mathcal{S}$. For any fixed $\mathcal{S}$, the decision rule can be wrong in two possible ways, false positive (Type I error) and false negative (Type II error). Mathematically, the Type I error rate $\mathcal{E}_{\mathrm{I}}(x) = \Pr[\mathcal{M}(x) \in \mathcal{S}]$ while the Type II error rate $\mathcal{E}_{\mathrm{II}}(x') = \Pr[\mathcal{M}(x') \notin \mathcal{S}] = 1 - \Pr[\mathcal{M}(x') \in \mathcal{S}]$. According to the definition of DP, for any pair of neighboring databases $x, x'$, the adversary always has

$$e^\epsilon \cdot \mathcal{E}_{\mathrm{I}}(x) + \mathcal{E}_{\mathrm{II}}(x') \geq 1 - \delta \quad \text{and} \quad e^\epsilon \cdot \mathcal{E}_{\mathrm{II}}(x') + \mathcal{E}_{\mathrm{I}}(x) \geq 1 - \delta,$$

which implies that $\mathcal{E}_{\mathrm{I}}(x)$ and $\mathcal{E}_{\mathrm{II}}(x')$ cannot be small at the same time. When $\epsilon$ and $\delta$ are both small, both $\mathcal{E}_{\mathrm{I}}$ and $\mathcal{E}_{\mathrm{II}}$ becomes close to 0.5 (the error rates of random guess), which means that the adversary cannot get much information from the output of $\mathcal{M}$.

**Justification 2: With high probability, $\mathcal{M}$ is insensitive to the change of one record (Guingona et al., 2023).**
In more detail, $(\epsilon, \delta)$-DP guarantees the distribution of $\mathcal{M}$'s output will not change significantly when replacing one record. According to Property (A) of Theorem 3.2 in Guingona et al. (2023), $(\epsilon, \delta)$-DP implies[3]

$$\Pr_{a \sim \mathcal{M}(x)} \left[ \frac{1}{2e^\epsilon} \leq \frac{\Pr[\mathcal{M}(x) = a]}{\Pr[\mathcal{M}(x') = a]} \leq 2e^\epsilon \right] \geq 1 - 2\delta \quad \text{for any pair of neighboring databases } x \text{ and } x'. \quad (2)$$

The above inequality shows that the change of one record cannot make an output significantly more likely or significantly less likely (with at least $1 - 2\delta$ probability). Since $x$ and $x'$ only differ in one record, the above formula also guarantees that the adversary cannot learn too much information about any single record of the database through observing the output of $\mathcal{M}$ (Dwork et al., 2014, p. 25).

**Justification 3: DP guarantees limited Bayesian adversary's utilities.**
We consider the same adversary as in Justification 1. Since the adversary has no information about the missing entry, he/she may assume a uniform prior distribution about the missing entry. To simplify notations, let $X_i \in \mathcal{X}$ denote the missing entry and let $x_{-i}$ denote the database $x$ with $X_i$ removed. For any $X_i \in \mathcal{X}$, the adversary's posterior distribution (after observing output $a$ from mechanism $\mathcal{M}$) is

$$\Pr[X_i | a, x_{-i}] = \frac{\Pr[a | X_i, x_{-i}] \cdot \Pr[X_i | x_{-i}]}{\Pr[a | x_{-i}]} = \frac{\Pr[\mathcal{M}(x_{-i} \cup \{X_i\}) = a] \cdot \Pr[X_i]}{\sum_{X'} \left( \Pr[\mathcal{M}(x_{-i} \cup \{X'\}) = a] \cdot \Pr[X'] \right)}$$
$$= \frac{\Pr[\mathcal{M}(x_{-i} \cup \{X_i\}) = a]}{\sum_{X'} \Pr[\mathcal{M}(x_{-i} \cup \{X'\}) = a]}.$$

A Bayesian predictor predicts the missing entry $X_i$ through maximizing the posterior probability. For the adversary with uniform prior, when the output is $a$, the 0/1 loss of the Bayesian predictor is

$$\ell_{0\text{-}1}(a, x_{-i}) = 0^2 \cdot \max_i \left( \Pr[X_i | a, x_{-i}] \right) + 1^2 \cdot \left( 1 - \max_i \left( \Pr[X_i | a, x_{-i}] \right) \right) = 1 - \max_i \left( \Pr[X_i | a, x_{-i}] \right).$$

---

[1]Some differentially private mechanisms such as "stability-based query release" appear to not require additive noise with high probability if the local sensitivity is 0 for the input database and for all other databases that differ by at most $\ln(1/\delta)/\epsilon$ data points (Thakurta & Smith, 2013). Note that in our case (such as the motivating example), the local sensitivity is not 0. Also, the "stability-based" methods still need randomization (though not adding noise).

[2]The adversary can use any decision rule, and the rejection region rule is adopted just for example.

[3]Theorem 3.2 in Guingona et al. (2023) shows $(\epsilon, \delta)$-DP implies $\left( (\ln 2)\epsilon, \ 2\delta \right)$-probabilistic DP, which is equivalent to (2). The formal definition of probabilistic DP can be found in Definition 6 in Machanavajjhala et al. (2008).

It's not hard to check that a always-correct prediction has zero loss and any always-incorrect prediction has loss one. Then, we define the adjusted utility of adversary (in Bayesian prediction), which is the expectation of a normalized version of $\ell_{0\text{-}1}$. Mathematically, for a database $x$, we define the adjusted utility with threshold $t$ as follows,

$$u(t, x_{-i}) = \frac{1}{1-t} \cdot \max_{X_i} \left( \mathbb{E}_{a \sim \mathcal{M}(x_{-i} \cup X_i)} \Big[ \max \big\{ 0, 1 - t - \ell_{0\text{-}1}(a) \big\} \Big] \right). \tag{3}$$

In short, $u(t, x_{-i})$ is the worst-case expectation of $1 - \ell_{0\text{-}1}$ while the contribution from predictors with loss larger than $1 - t$ is omitted. Especially, when the threshold $t \geq 1/|\mathcal{X}|$, an always correct predictor ($\ell_{0\text{-}1} = 0$) has utility 1 and a random guess predictor ($\ell_{0\text{-}1} = 1 - 1/|\mathcal{X}|$) has utility 0. For example, we let $\mathcal{X} = \{0, 1\}$ and consider the coin-flipping mechanism $\mathcal{M}_{\text{CF}}$ with support $\mathcal{X}$, which output $X_i$ with probability $p$ and output $1 - X_i$ with probability $1 - p$. When $p = 1$, the entry $X_i$ is non-private because the adversary can directly learn it from the output of $\mathcal{M}_{\text{CF}}$. Correspondingly, the adjusted utility of adversary is 1 for any threshold $t \in (0, 1)$. When $p = 0.5$, the mechanism gives an output uniformly at random from $\mathcal{X}$. In this case, the output of $\mathcal{M}$ cannot provide any information to the adversary. Correspondingly, the adjusted utility of adversary is 0 for any threshold $t \in (0.5, 1)$. The following lemma, which is a direct corollary of Theorem 1 and Corollary 3, shows that the adjusted utility is upper bounded by the $\delta$ parameter of DP. Note that Lemma 1 also matches the main message of Figure 1.

**Lemma 1.** *Let mechanism $\mathcal{M}$ be $(\epsilon, \delta)$-differentially private. Then, for the above-defined adjusted utility,*

$$u\Big( \frac{e^\epsilon}{e^\epsilon + 1}, \, x_{-i} \Big) \leq 2\delta.$$

In other words, when both $\epsilon$ and $\delta$ are small, the adversary cannot accurately predict the missing entry in database $x$.

## 3 Smoothed Differential Privacy

Recall that DP is based on the worst-case analysis over all possible databases. However, as shown in Example 1 and Figure 1, the worst-case nature of DP sometimes leads to an overly pessimistic measure of privacy loss, which may bring unnecessary additive noise in the hope of improved privacy but at the cost of accuracy. For example, we assume a database with $n$ binary records. DP considers the worst-case of $x$. Here, we focus on one of the worst-cases, $x = (0, \cdots, 0, 1)$, whose worst-case neighboring database $x' = (0, \cdots, 0, 0)$. From the adversaries' point of view, he/she knows all records in the database except the last one (the first $n - 1$ records). Under the above worst-case, the sampling-histogram mechanism is non-private because the adversary directly knows the missing entry if the last entry, "1", got sampled. However, the above worst-case might be very rare in some real-world applications (*e.g.*, Example 1) and the mechanism is actually private when the database is not extremely close to the worst case (Figure 1). Motivated by this, we propose smoothed DP, which has similar privacy guarantees as DP, but is able to measure the privacy level of sampling-based mechanisms (instead of classifying all of them as non-private). This section formally introduces smoothed DP, which applies the smoothed analysis to the database-dependent privacy profile $\delta(x)$ and proves its desirable properties. Due to the space constraint, all proofs of this section can be found in Appendix C.

### 3.1 The database-dependent privacy profile

We first introduce the *database-dependent privacy profile* $\delta_{\epsilon, \mathcal{M}}(x)$, which measures the privacy leakage of mechanism $\mathcal{M}$ when its input is $x$. Here, we fix $\epsilon$ and let $\delta$ be database-dependent, which is opposite to data-dependent privacy loss (Papernot et al., 2018; Wang, 2019) where $\delta$ is fixed and $\epsilon$ is data-dependent. Here, we call $\delta$ *privacy profile* since it is a function of $\epsilon$ (Balle et al., 2020).

**Definition 2** (**Database-dependent privacy profile $\delta_{\epsilon, \mathcal{M}}(x)$** (Dwork et al., 2006b)). *Let $\mathcal{M} : \mathcal{X}^n \to \mathcal{A}$ denote a randomized mechanism. Given any database $x \in \mathcal{X}^n$ and any $\epsilon > 0$, define the database-dependent privacy profile as:*

$$\delta_{\epsilon, \mathcal{M}}(x) \triangleq \max \Big( 0, \, \max_{x' : x' \simeq x} \big( d_{\epsilon, \mathcal{M}}(x, x') \big), \, \max_{x' : x' \simeq x} \big( d_{\epsilon, \mathcal{M}}(x', x) \big) \Big),$$

*where $d_{\epsilon, \mathcal{M}}(x, x') = \max_{\mathcal{S}} \big( \Pr\left[ \mathcal{M}(x) \in \mathcal{S} \right] - e^\epsilon \cdot \Pr\left[ \mathcal{M}(x') \in \mathcal{S} \right] \big)$ and "$\simeq$" means neighboring.*

In words, $\delta_{\epsilon, \mathcal{M}}(x)$ is the minimum $\delta$ values, such that the $(\epsilon, \delta)$-DP requirement on $\mathcal{M}$ (Inequality (1)) holds for any neighboring pairs $(x, x')$ and $(x', x)$. The definition of $\delta_{\epsilon, \mathcal{M}}(x)$ guarantees the adversaries to have the same prior

knowledge as DP. $\delta_{\epsilon,\mathcal{M}}(x)$ considers the worst-case neighboring dataset $x'$ (technically, $\max_{x \simeq x'}$), which indicates that the adversaries know the whole database except one entry.

The next lemma reveals the connection between the adversary's utility $u$ (defined in Justification 3 of Section 2) and $d_{\epsilon,\mathcal{M}}(x, x') + d_{\epsilon,\mathcal{M}}(x', x)$.

**Lemma 2.** *Given mechanism $\mathcal{M} : \mathcal{X}^n \to \mathcal{A}$ and any pair of neighboring databases $x \simeq x'$,*

$$u\Big(\frac{e^\epsilon}{e^\epsilon + 1}, x \cap x'\Big) < d_{\epsilon,\mathcal{M}}(x, x') + d_{\epsilon,\mathcal{M}}(x', x).$$

Lemma 2 shows that the adjusted utility is upper bounded by $d_{\epsilon,\mathcal{M}}$. Especially, when $|\mathcal{X}| = 2$, we provide both upper and lower bounds to the adjusted utility in Lemma 10 of Appendix C.1, which means that $d_{\epsilon,\mathcal{M}}(x, x')$ is a good measure for the privacy level of $\mathcal{M}$ when $|\mathcal{X}| = 2$. The following corollary shows that $\delta_{\epsilon,\mathcal{M}}(x)$ upper bounds the adjusted utility of adversary. In other words, a small $\delta_{\epsilon,\mathcal{M}}(x)$ guarantees that adversary cannot accurately predict the missing record in the database.

**Corollary 3.** *Given mechanism $\mathcal{M} : \mathcal{X}^n \to \mathcal{A}$ and any pair of neighboring databases $x \simeq x'$,*

$$u\Big(\frac{e^\epsilon}{e^\epsilon + 1}, x \cap x'\Big) < 2 \cdot \delta_{\epsilon,\mathcal{M}}(x).$$

**DP as the worst-case analysis of $\delta_{\epsilon,\mathcal{M}}(x)$.** In the next theorem, we show that the privacy measure based on the worst-case analysis of $\delta_{\epsilon,\mathcal{M}}$ is equivalent to the standard DP.

**Theorem 1 (DP in $\delta_{\epsilon,\mathcal{M}}(x)$).** *Mechanism $\mathcal{M} : \mathcal{X}^n \to \mathcal{A}$ is $(\epsilon, \delta)$-differentially private if and only if,*

$$\max_{x \in \mathcal{X}^n} \Big(\delta_{\epsilon,\mathcal{M}}(x)\Big) \le \delta.$$

### 3.2 Formal definition of smoothed DP

Armed with the database-dependent privacy profile $\delta_{\epsilon,\mathcal{M}}(x)$, we now formally define smoothed DP, where the worst-case "ground truth" distribution of every agent is allowed to be any distribution from a set of distributions $\Pi$, on top of which Nature adds random noises to generate the database. We would like to note again that $\Pi$ is a parameter for smoothed analysis instead of the mechanisms $\mathcal{M}$. The smoothed analysis only controls how the database $x$ is generated. The adversaries of smoothed DP will have the same prior knowledge as the adversaries of DP.

**Definition 3 (Smoothed DP).** *Let $\Pi$ be a set of distributions over $\mathcal{X}$. We say $\mathcal{M} : \mathcal{X}^n \to \mathcal{A}$ is $(\epsilon, \delta, \Pi)$-smoothed differentially private if,*

$$\max_{\vec{\pi} \in \Pi^n} \big( \mathbb{E}_{x \sim \vec{\pi}} [\delta_{\epsilon,\mathcal{M}}(x)] \big) \le \delta,$$

*where $x \sim \vec{\pi} = (\pi_1, \cdots, \pi_n)$ means that the $i$-th entry in the database follows $\pi_i$ for every $i \in \{1, \cdots, n\}$.*

**The threat models of DP, smoothed DP, and distributional DP** are shown in Table 3. Note that the adversary "is able to choose" means that he/she can select the worst-case, and does not necessarily mean the adversary knows the information. In short, the adversary in smoothed DP is as knowledgeable as that in DP, but with less ability to choose the database $x$. Compared with distributional DP, the adversary in smoothed DP has more information (prior knowledge) about the database. Appendix B rigidly shows that smoothed DP is a stronger notion than distributional DP.

| Adversary's | Prior knowledge | Ability to choose database $x$ | Ability to choose neighboring database $x'$ | Ability to choose output $\mathcal{S}$ |
|---|---|---|---|---|
| DP | **Whole database except one entry** ($x \cap x'$) | Able to choose the worst-case database $x$ | **Able to choose worst-case** diff$(x, x')$ | **Able to choose worst-case output $\mathcal{S}$** |
| **Smoothed DP** | | **Able to choose data distributions from $\Pi$** | | |
| Distributional DP | The distribution of $x \cap x'$ | | | |

Table 3: The threat model of DP, smoothed DP, and distribution DP, where diff$(x, x')$ represents the difference between $x$ and $x'$. Under the setting and notation of Justification 3 of DP, diff$(x, x')$ is $X_i$ while $x \cap x'$ is $x_{-i}$.

Like DP, smoothed DP bounds privacy leakage (in an arguably more realistic setting), via the following three justifications that are similar to the two common justifications of DP in Section 2.

**Justification 1: Smoothed DP prevents membership inference.** Mathematically, a $(\epsilon, \delta, \Pi)$-smoothed DP mechanism $\mathcal{M}$ guarantees

$$e^\epsilon \cdot \max_{\vec{\pi} \in \Pi^n} \big( \mathop{\mathbb{E}}_{x \sim \vec{\pi}} [\mathcal{E}_{\mathrm{I}}(x)] \big) + \max_{\vec{\pi} \in \Pi^n} \big( \mathop{\mathbb{E}}_{x \sim \vec{\pi}} [\mathcal{E}_{\mathrm{II}}(x') \,|\, x' \simeq x] \big) \geq 1 - \delta$$

$$e^\epsilon \cdot \max_{\vec{\pi} \in \Pi^n} \big( \mathop{\mathbb{E}}_{x \sim \vec{\pi}} [\mathcal{E}_{\mathrm{II}}(x') \,|\, x' \simeq x] \big) + \max_{\vec{\pi} \in \Pi^n} \big( \mathop{\mathbb{E}}_{x \sim \vec{\pi}} [\mathcal{E}_{\mathrm{I}}(x)] \big) \geq 1 - \delta$$

The proof follows after bounding Type I and Type II errors when the input is $x$ by $\delta_{\epsilon, \mathcal{M}}$. That is, for a fixed database $x$, it is not hard to verify that

$$e^\epsilon \cdot \mathcal{E}_{\mathrm{I}}(x) + \mathcal{E}_{\mathrm{II}}(x') \geq 1 - \max_{x' : x' \simeq x} \big( d_{\epsilon, \mathcal{M}}(x, x') \big) \quad \text{and}$$

$$e^\epsilon \cdot \mathcal{E}_{\mathrm{II}}(x') + \mathcal{E}_{\mathrm{I}}(x) \geq 1 - \max_{x' : x' \simeq x} \big( d_{\epsilon, \mathcal{M}}(x', x) \big)$$

Then, by the definition $\delta_{\epsilon, \mathcal{M}}(x)$, we have,

$$e^\epsilon \cdot \mathcal{E}_{\mathrm{I}}(x) + \mathcal{E}_{\mathrm{II}}(x') \geq 1 - \delta_{\epsilon, \mathcal{M}}(x) \quad \text{and} \quad e^\epsilon \cdot \mathcal{E}_{\mathrm{II}}(x') + \mathcal{E}_{\mathrm{I}}(x) \geq 1 - \delta_{\epsilon, \mathcal{M}}(x),$$

which means that $\mathcal{E}_{\mathrm{I}}$ and $\mathcal{E}_{\mathrm{II}}$ cannot be small at the same time when the database is $x$. Then, justification 1 can be gotten by applying smoothed analysis to both sides. It follows that the smoothed DP, which is a smoothed analysis of $\delta_{\epsilon, \mathcal{M}}$, can bound the smoothed Type I and Type II errors.

**Justification 2: Smoothed DP mechanisms are insensitive to the change of one record with high probability.** Mathematically, a $(\epsilon, \delta, \Pi)$-smoothed DP mechanism $\mathcal{M}$ guarantees

$$\max_{\vec{\pi} \in \Pi^n} \left( \mathop{\mathbb{E}}_{x \sim \vec{\pi}} \left[ \Pr_{a \sim \mathcal{M}(x)} \left[ \frac{1}{2e^\epsilon} \leq \frac{\Pr[\mathcal{M}(x) = a]}{\Pr[\mathcal{M}(x') = a]} \leq 2e^\epsilon \right] \right] \right) \geq 1 - 2\delta$$

The proof is, again, done through analyzing $\delta_{\epsilon, \mathcal{M}}(x)$. More precisely, given any mechanism $\mathcal{M}$, any $\epsilon \in \mathbb{R}_+$ and any pair of neighboring databases $x, x'$, we have

$$\Pr_{a \sim \mathcal{M}(x)} \left[ \frac{1}{2e^\epsilon} \leq \frac{\Pr[\mathcal{M}(x) = a]}{\Pr[\mathcal{M}(x') = a]} \leq 2e^\epsilon \right] \geq 1 - 2\delta_{\epsilon, \mathcal{M}}(x).$$

Then, justification 2 follows by applying smoothed analysis to both sides of the above inequality.

As smoothed DP replaces the worst-case analysis with smoothed analysis, we also view $\delta = o(1/n)$ as a requirement for private mechanisms for smoothed DP. In addition to the two justifications above,

**Justification 3: Smoothed DP guarantees limited adversaries' utility in (Bayesian) predictions.**
Following similar reasoning as Justification 1, we know that the utility under realistic settings (or the smoothed utility) of the adversary cannot be larger than $2\delta$. Mathematically, an $(\epsilon, \delta, \Pi)$-smoothed DP mechanism $\mathcal{M}$ guarantees

$$\max_{\vec{\pi} \in \Pi^n} \left( \mathbb{E}_{x \sim \vec{\pi}} \left[ u\Big( \frac{e^\epsilon}{e^\epsilon + 1}, x \cap x' \Big) \right] \right) < 2\delta.$$

At a high level, a small $\delta$ parameter of smoothed DP means the adversary cannot accurately predict the missing entry of database under realistic settings.

## 4 Properties of Smoothed DP

First, we present four properties of smoothed DP and discuss how they can help mechanism designers figure out the smoothed DP parameters of mechanisms. We first present the robustness to *post-processing* property, which says no function can make a mechanism less private without adding extra knowledge about the database. The post-processing property of smoothed DP can be used to upper bound the privacy level of many mechanisms. With it, we know private data preprocessing can guarantee the privacy of the whole mechanism. Then, the rest part of the mechanism does not need to consider privacy issues. The proof of all four properties of the smoothed DP can be found in Appendix D.

**Proposition 4** (**Post-processing**). *Let $\mathcal{M} : \mathcal{X}^n \to \mathcal{A}$ be a $(\epsilon, \delta, \Pi)$-smoothed DP mechanism. For any $f : \mathcal{A} \to \mathcal{A}'$ (which can also be randomized), $f \circ \mathcal{M} : \mathcal{X}^n \to \mathcal{A}'$ is also $(\epsilon, \delta, \Pi)$-smoothed DP.*

Then, we introduce the composition theorem for the smoothed DP, which bounds the smoothed DP property of databases when two or more mechanisms publish their outputs about the same database.

**Proposition 5** (**Composition**). *Let $\mathcal{M}_i : \mathcal{X}^n \to \mathcal{A}_i$ be an $(\epsilon_i, \delta_i, \Pi)$-smoothed DP mechanism for any $i \in [k]$. Define $\mathcal{M}_{[k]} : \mathcal{X}^n \to \prod_{i=1}^k \mathcal{A}_i$ as $\mathcal{M}_{[k]}(x) = \big(\mathcal{M}_1(x), \cdots, \mathcal{M}_k(x)\big)$. Then, $\mathcal{M}_{[k]}$ is $\left(\sum_{i=1}^k \epsilon_i, \sum_{i=1}^k \delta_i, \Pi\right)$-smoothed DP.*

In practice, $\Pi$ might be hard to accurately characterize. The next proposition introduces the pre-processing property of smoothed DP, which says the distribution of data ($\Pi$) can be replaced by the distribution of features ($f(\Pi)$, defined as follows) if the features are extracted using any deterministic function(s). Note that this only replaces the smoothed analysis-related $\Pi$ and the mechanism remains unchanged. For example, in deep learning, the distribution of data can be replaced by the distribution of gradients[4], which is usually much easier to estimate in real-world training processes.

More technically, the pre-processing property guarantees that any deterministic way of data preprocessing is not harmful to privacy. To simplify notation, we let $f(\pi)$ be the distribution of $f(X)$ where $X \sim \pi$. For any set of distributions $\Pi = \{\pi_1, \cdots, \pi_m\}$, we let $f(\Pi) = \{f(\pi_1), \cdots, f(\pi_m)\}$.

**Proposition 6** (**Pre-processing for deterministic functions**). *Let $f : \mathcal{X}^n \to \tilde{\mathcal{X}}^n$ be a deterministic function and $\mathcal{M} : \tilde{\mathcal{X}}^n \to \mathcal{A}$ be a randomized mechanism. Then, $\mathcal{M} \circ f : \mathcal{X}^n \to \mathcal{A}$ is $(\epsilon, \delta, \Pi)$-smoothed DP if $\mathcal{M}$ is $\big(\epsilon, \delta, f(\Pi)\big)$-smoothed DP.*

The following proposition shows that any two sets of distributions with the same convex hull have the same privacy level under smoothed DP. With this, the mechanism designers can ignore all inner points and only consider the convex hull's vertices when calculating the mechanisms' privacy level. Let $\mathrm{CH}(\Pi)$ denote the convex hull of $\Pi$.

**Proposition 7** (**Distribution reduction**). *Given any $\epsilon, \delta \in \mathbb{R}_+$ and any $\Pi_1$ and $\Pi_2$ such that $\mathrm{CH}(\Pi_1) = \mathrm{CH}(\Pi_2)$, a mechanism $\mathcal{M}$ is $(\epsilon, \delta, \Pi_1)$-smoothed DP if and only if $\mathcal{M}$ is $(\epsilon, \delta, \Pi_2)$-smoothed DP.*

We provide an example of how Proposition 7 helps calculate the privacy level. Assume the database has two possible types of data (*i.e.*, $m \triangleq |\mathcal{X}| = 2$). We use $\pi = (1, 1-p)$ to represent the distribution over $|\mathcal{X}|$ such that the first type occurs with probability $p$ and the second type occurs with probability $1-p$. Assuming the mechanism designer considers an infinite set of distribution $\Pi = \big\{(p, 1-p) : p \in (0.2, 0.8)\big\}$, its easy to check that $\Pi$'s convex hull $\mathrm{CH}(\Pi) = \big\{(p, 1-p) : p \in [0.2, 0.8]\big\}$ and $\mathrm{CH}(\Pi)$'s set of vertices $\Pi^* = \big\{(p, 1-p) : p \in \{0.2, 0.8\}\big\} = \big\{(0.2, 0.8), (0.8, 0.2)\big\}$. Since $\Pi^*$ and $\Pi$ have the same convex hull, according to Proposition 7, the mechanism designer only needs to consider $\Pi^*$ (two distributions), instead of the infinite set $\Pi$.

Proposition 7 also provides an efficient way to calculate the (exact) $\delta$ parameter of smoothed DP for anonymous mechanisms. Here, "anonymous" means the mechanism treat each data in the database "equally". Formally, we say one mechanism is anonymous if its output distribution will not change under arbitrarily permuted order of data in the database. In other words, an anonymous mechanism's output distribution only depends on the histogram of data. One can see that most commonly-used algorithms in machine learning (*e.g.*, SGD and AdaGrad) and voting (*e.g.*, plurality, Borda, and Copeland) are anonymous. Also note that the commonly used noisy mechanisms (*e.g.*, Laplacian, Gaussian, and Exponential) will keep the anonymity property of mechanisms.

Algorithm 1 efficiently calculate the privacy profile $\delta$ of smoothed DP for anonymous mechanisms. Again, we let $\Pi^*$ denote the set of $\mathrm{CH}(\Pi)$'s vertices and let $\ell^* \triangleq |\Pi^*|$ denote the cardinality of $\Pi^*$. $\mathcal{Q}$ denote the set of all histograms on $\Pi^*$ of $n$ records. Here, we slightly abuse notation and use "**for** $\vec{\pi} \in \mathcal{Q}$" to represent that each histogram is visited exactly once in the corresponding for loop.

---

**Algorithm 1:** Calculate the (exact) privacy profile $\delta$ for smoothed DP

**1 Inputs:** An anonymous mechanism $\mathcal{M}$, parameter $\epsilon \in \mathbb{R}_+$, size of database $n$, and a set of distribution $\Pi$

**2 Initialization:** Calculate $\Pi^*$ according to $\Pi$ and $\mathcal{Q}$ according to $\Pi^*$

**3** (Same as DP) Calculate $\delta_{\epsilon, \mathcal{M}}$ for all histograms

**4 for** $\vec{\pi} \in \mathcal{Q}$ **do**

**5**     Calculate $\delta_j \triangleq \mathbb{E}_{x \sim \vec{\pi}}[\delta_{\epsilon, \mathcal{M}}(x)]$

**6 end**

**7 Output:** $\delta_{\mathrm{SDP}}(\epsilon) = \max_j \delta_j$

---

[4]Gradient is what the mechanism designers target to privatize for deep neural networks. This is because one of the most serious privacy leakage in deep learning is the gradient residue on cloud GPUs (its VRAM can be visible to other cloud users, see Abadi et al., 2016). We call gradients as features here because it is calculated according to the data in the database.

We say Step 3 in Algorithm 1 is the "same as" DP because calculating the privacy profile $\delta$ of DP for general mechanisms requires calculating $\delta_{\epsilon,\mathcal{M}}$. We note calculating the exact privacy profile $\delta$ of DP may not require calculating $\delta_{\epsilon,\mathcal{M}}$ for some mechanisms (*e.g.*, additive noise mechanism). We will present a similar result for smoothed DP in Theorem 3 of Section 5.1. Theorem 2 presents the runtime of Algorithm 1 (calculate the privacy profile $\delta$ of smoothed DP for any anonymous mechanisms), which only requires a polynomial extra time than DP in most cases.

**Theorem 2** (Runtime of Algorithm 1). *Using the notations above, the complexity of calculating the privacy profile $\delta$ of smoothed DP for any anonymous mechanisms is $O\left(n^{2m+\ell^*-3} + \ell \log \ell^*\right) + \mathrm{Cpl}_{\mathrm{DP}}$, where $\ell$ denotes the number of vertices in $\Pi$ and $\mathrm{Cpl}_{\mathrm{DP}}$ represents the runtime of Step 3 in Algorithm 1.*

As to be shown in Section 5, the application scenarios of Smoothed DP are those when $m$ is not large. $\ell$ could be treated as a constant if $\Pi$'s geometric shape is not extremely complicated. Thus, we believe the runtime of calculating the exact privacy profile $\delta$ only requires a polynomial extra time than DP in most cases.

## 5 Smoothed DP as a Measure of Privacy

This section uses smoothed DP to measure the privacy of some commonly-used mechanisms, where the sampling noise is intrinsic and unavoidable (as opposed to additive noises such as Gaussian or Laplacian noises). Our analysis focuses on two widely-used algorithms where the intrinsic randomness comes from sampling without replacement. In addition, we compare the privacy levels of smoothed DP with DP. All missing proofs of this section can be found in Appendix E.

### 5.1 Discrete mechanisms are more private than what DP predicts

In this subsection, we study the smoothed DP property of (discrete) sampling-histogram mechanism (SHM), which is widely used as a pre-possessing step in many real-world applications like the training of NNs. As smoothed DP satisfies *post-processing* (Proposition 4) and *pre-processing* (Proposition 5), the smoothed DP property of SHM can upper bound the privacy level of many mechanisms (that uses SHM) in practice. Let $\mathcal{X}$ denote a finite set and $m \triangleq |\mathcal{X}|$ denotes the cardinality of $\mathcal{X}$. The histogram of a database $x \in \mathcal{X}^n$ (denoted as **hist**$(x)$) is an $m$-dimensional vector. The $j$-th component of **hist**$(x)$ is the number of $j$-th type of data in $x$. It's easy to check that **hist**$(x) \in \{0, \cdots, n\}^m$ and $\|\mathbf{hist}(x)\|_1 = n$.

<table>
<tr><td>

SHM first sample $T = \lceil \eta \cdot n \rceil$ data from the database and then output the histogram of the $T$ samples. Formally, we define the sampling-histogram mechanism in Algorithm 2. Note that we require all data in the database to be chosen from a finite set $\mathcal{X}$.

</td><td>

**Algorithm 2:** Sampling-histogram mechanism $\mathcal{M}_H$

**1 Inputs:** A finite set $\mathcal{X}$, sampling rate $\eta \in (0,1)$, and a database $x = \{X_1, \cdots, X_n\}$ where $X_i \in \mathcal{X}$ for all $i \in \{1, \cdots, n\}$
**2** Randomly sample $T = \lceil \eta \cdot n \rceil$ data from $x$ without replacement. The sampled data are denoted by $X_{j_1}, \cdots, X_{j_T}$.
**3 Output:** The histogram **hist**$(X_{j_1}, \cdots, X_{j_T})$

</td></tr>
</table>

**Smoothed DP of mechanisms based on SHM.** The smoothed DP of SHM can be used to upper bound the smoothed DP of the following three groups of mechanisms/algorithms.

The first group consists of deterministic voting rules, as presented in the motivating example in Introduction. The sampling procedure in SHM mimics the votes that got lost.

The second group consists of machine learning algorithms based on randomly-sampled training data, such as cross-validation. The (random) selection of the training data corresponds to SHM. Notice that many training algorithms are essentially based on the histogram of the training data (instead of the ordering of data points). Therefore, overall the training procedure can be viewed as SHM plus a post-processing function (the learning algorithm). Consequently, the smoothed DP of SHM can be used to upper bound the smoothed DP of such procedures.

The third group consists of SGD of NNs with gradient quantization (Zhu et al., 2020; Banner et al., 2018), where the gradients are rounded to 8-bit in order to accelerate the training and/or the inference of NNs. The smoothed DP of SHM can be used to bound the privacy leakage in each SGD step of the NN, where a batch (a subset of the training set) is firstly sampled and the gradient is the average of the gradients of the sampled data.

**DP vs. Smoothed DP for SHM.** We are ready to present the main theorem of this paper, which indicates that SHM is private under some mild assumptions. We say distribution $\pi$ is $f$-*strictly positive* if there exists a positive function

$f(n,m)$ such that $\pi(X) \geq f(n,m)$ for any $X$ in the support of $\pi$. A set of distributions $\Pi$ is $f$-*strictly positive* if there exists a positive function $f(n,m)$ such that every $\pi \in \Pi$ is $f$-strictly positive. The $f$-strictly positive assumption is often considered mild in *elections* (Xia, 2020) and *discrete machine learning* (Laird & Saul, Apr. 1994) when $f(m,n) = O(1)$. All following discussions in this section assume $m$ to be a constant. Note that constants are omitted when analyzing asymptotic manner (*e.g.*, as a multiple within $O(\cdot)$, $\Theta(\cdot)$, or $\omega(\cdot)$).

**Theorem 3** (**DP vs. Smoothed DP for Sampling Histogram Mechanism $\mathcal{M}_H$**). *For any $\mathcal{M}_H$, given an $f$-strictly positive set of distributions $\Pi$, a finite set $\mathcal{X}$, and $n, T \in \mathbb{Z}_+$, we have:*

$(i)$ (**Smoothed DP**) *Given any $\epsilon \geq \ln\left(\frac{1}{1-\eta}\right) + c$ where $c$ is a constant, $\mathcal{M}_H$ is $\left(\epsilon, \, m \cdot \exp\left[-\Theta\big(f(n,m) \cdot n\big)\right], \, \Pi\right)$-smoothed DP.*

$(ii)$ (**Tightness of smoothed DP bound**) *For any $\epsilon > 0$, there does not exist $\delta = \exp\left(-\omega\big([\ln f(m,n)] \cdot n\big)\right)$ such that $\mathcal{M}_H$ is $(\epsilon, \delta, \Pi)$-smoothed DP.*

$(iii)$ (**DP**) *For any $\epsilon > 0$, there does not exist $\delta < \eta$ such that $\mathcal{M}_H$ is $(\epsilon, \delta)$-DP.*

This theorem says the privacy leakage is exponentially small under real-world application scenarios. In comparison, DP cares too much about the extremely rare cases and predicts a constant privacy leakage. If $f$ is a constant function and $m = \Theta(1)$, Property $(ii)$ indicates that the bound in $(i)$ is tight. Also, note that our theorem allows $T$ to be in the same order as $n$. For example, when setting $T = 95\% \times n$ and $\Pi$ be $f$-strictly positive for a constant $f$, SHM is $(3, \exp(-\Theta(n)), \Pi)$-smoothed DP, which is an acceptable privacy threshold in many real-world applications (Liu et al., 2019). For example, iOS 10.12.3 requires $\epsilon \leq 6$ and iOS 10.1.1 requires $\epsilon \leq 14$ Tang et al. (2017). Appendix F proves similar bounds for the SHM with replacement. The following remarks shows a non-asymptotic version of Theorem 3$(i)$, which relates $\epsilon$ and $\delta$ and provides a non-asymptotic privacy bound for SHM. To simplify notations, we let $g(\eta, \epsilon) \triangleq \left(\eta^{-1}(1 - e^{-\epsilon}) - 1\right)^2$.

**Remark 8** (Privacy bound for $\mathcal{M}_H$). *Given an $f$-strictly positive set of distributions $\Pi$, a finite set $\mathcal{X}$, and $n, T \in \mathbb{Z}_+$, $\mathcal{M}_H$ is $\left(\epsilon, \, \exp\left(-\frac{1}{6} \cdot g(\eta, \epsilon) \cdot f(m,n) \cdot n\right) + m \cdot \exp\left(-\frac{1}{8} \cdot f(m,n) \cdot n\right), \, \Pi\right)$-smoothed DP for any $\epsilon > \ln\left(\frac{1}{1-\eta}\right)$.*

## 5.2 Continuous mechanisms are similar to what DP predicts

In this section, we show that the sampling mechanisms with continuous support are still not privacy-preserving under smoothed DP. The "gap" between discrete and continuous SHM comes from the fact that SHM becomes less private when $|\mathcal{X}|$ increases.

---
**Algorithm 3:** Continuous sampling-average mechanism $\mathcal{M}_A$

---
1 **Inputs:** The number of samples $T$ and a database $x = \{X_1, \cdots, X_n\}$ where $X_i \in [0,1]$ for all $i \in \{1, \cdots, n\}$
2 Randomly sample $T = \lceil \eta \cdot n \rceil$ data from $x$ without replacement. The sampled data are denoted as $X_{j_1}, \cdots, X_{j_T}$.
3 **Output:** The average $\bar{x} = \frac{1}{T} \sum_{j \in [T]} X_{i_j}$

---

Our result indicates that the neural networks without quantized parameters are not private without additive noise (*e.g.*, Gaussian or Laplacian noise). We use the sampling-average (Algorithm 3) algorithm as the standard algorithm for continuous mechanisms. Because sampling-average can be treated as SHM plus an average step, sampling-average is non-private also means SHM with continuous support is also non-private according to the post-processing property of smoothed DP.

**Theorem 4** (**Smoothed DP for continuous sampling-average**). *For any continuous sampling-average mechanism $\mathcal{M}_A$, given any set of strictly positive[5] distribution $\Pi$ over $[0,1]$, any $T, n \in \mathbb{Z}_+$ and any $\epsilon \geq 0$, there does not exist $\delta < \eta$ such that $\mathcal{M}_A$ is $(\epsilon, \delta, \Pi)$-smoothed DP.*

Theorem 4 does not contradict Property $(i)$ of Theorem 3 because the continuous functions has $m \to \infty$, which makes the upper bound of $\delta$ parameter for smoothed DP $m \cdot \exp\left[-\Theta\big(f(n,m) \cdot n\big)\right] \to \infty$.

---

[5]Distribution $\pi$ is strictly positive by $c$ if $p_\pi(x) \geq c$ for any $x$ in the support of $\pi$, where $p_\pi$ is the PDF of $\pi$.

## 6 Experiments

**Smoothed DP in elections.** We use a similar setting as the motivating example, where $0.2\%$ of the votes are randomly lost. We numerically calculate the (exact) privacy profile $\delta$ of smoothed DP according to Algorithm 1. Here, the set of distributions $\Pi$ includes the distribution of all 57 congressional districts of the 2020 presidential election. Using the *distribution reduction* property of smoothed DP (Proposition 7), we can remove all distributions in $\Pi$ except DC and NE-2[6], which are the vertices for the convex hull of $\Pi$. Figure 2 (left) shows that the smoothed $\delta$ parameter is exponentially small in $n$ when $\epsilon = 7$, which matches our Theorem 3. We find that $\delta$ is also exponentially small when $\epsilon = 0.5, 1$ or $2$, which indicates that the sampling-histogram mechanism is also more private than DP's predictions for small $\epsilon$'s. Appendix G.2 shows the experiments with different settings on $\Pi$. All experiments of this paper are implemented in MATLAB 2021a and tested on a Windows 10 Desktop with an Intel Core i7-8700 CPU and 32GB RAM.

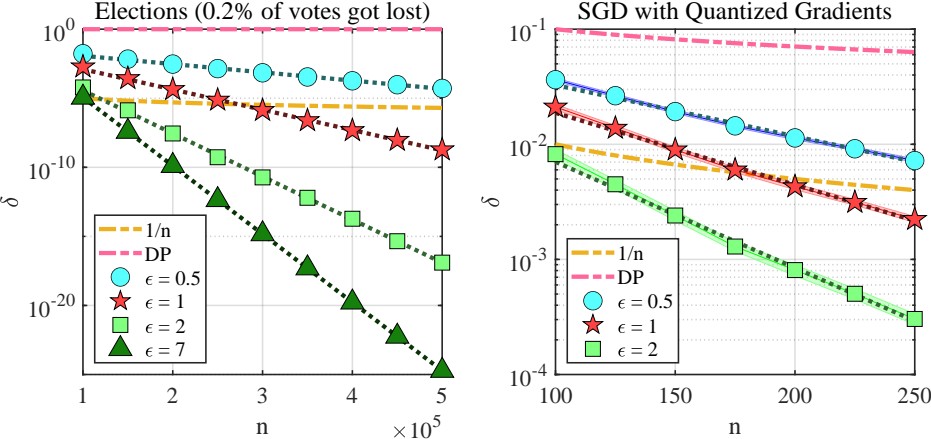

Figure 2: Left: DP and smoothed DP of 2020 US presidential election when $0.2\%$ of votes got lost. Right: DP and smoothed DP of 1-step SGD with 8-bit gradient quantization when the set of distributions is $\Pi$. In both plots, the vertical axes are in log-scale and the pink dashed line presents the $\delta$ parameter of DP with whatever (finite) $\epsilon$. The dot lines are the exponential fittings of smoothed $\delta$ parameters. The left plot is an accurate calculation of $\delta$. The shaded area shows the 99% confidence interval of the right plot.

**SGD with 8-bit gradient quantization.** According to the pre-processing property of smoothed DP, the smoothed DP of (discrete) sampling-average mechanism upper bounds the smoothed DP of SGD (for one step). In 8-bit neural networks for computer vision tasks, the gradient usually follows Gaussian distributions (Banner et al., 2018). We thus let the set of distributions $\Pi = \{\mathcal{N}_{\text{8-bit}}(0, 0.12^2), \mathcal{N}_{\text{8-bit}}(0.2, 0.12^2)\}$, where $\mathcal{N}_{\text{8-bit}}(\mu, \sigma^2)$ denotes the 8-bit quantized Gaussian distribution (See Appendix G for its formal definition). The standard variation, $0.12$, is the same as the standard variation of gradients in a ResNet-18 network trained on CIFAR-10 database (Banner et al., 2018). We use the standard setting of batch size $T = \sqrt{n}$. Figure 2 (right) shows that the smoothed $\delta$ parameter is exponentially small in $n$ for the SGD with 8-bit gradient quantization. The probabilities are estimated through $10^6$ independent trails. This result implies that the neural networks trained through quantized gradients can be private without adding additive noises. Also, see Appendix G.2 for the experiments under another three different settings on $\Pi$.

## 7 Conclusions and Future Work

We propose a novel notion to measure the privacy leakage of mechanisms without additive noises under realistic settings. One promising next step is to apply our smoothed DP notion to the entire training process of quantized NNs. Is the quantized NN private without additive noise? If not, what level of additive noises needs to be added, and how should we add noises in an optimal way? More generally, we believe that our work has the potential of making many algorithms private without requiring too much additive noise.

---

[6]DC refers to Washington, D.C., and NE-2 refers to Nebraska's 2nd congressional district.

## Acknowledgments

We thank the anonymous reviewers for their helpful comments. Lirong Xia acknowledges NSF #1453542, #2007994, #2106983, and a gift fund from Google. Yu-Xiang Wang's work on this project was partially supported by NSF Award #2048091.

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
