# OpenReview forum: "Smoothed Differential Privacy"
_TMLR — Accepted by TMLR_

### Review · Reviewer_zTAA · 2023-08-26

**Summary Of Contributions:**

The paper is proposing a variant of differential privacy (DP), where instead of using worst-case estimates, one can use a "smoothed" version of differential privacy. This allows the authors to argue about privacy leakage in a more realistic sense.

Along these lines the authors prove that any discrete mechanism with sampling procedures is more private that what standard differential privacy predicts. Furthermore, the authors prove that many continuous mechanisms with sampling procedures are still non-private under smoothed differential privacy.

In addition, the authors prove some desirable properties of smoothed differential privacy, such as composition, robustness to post-processing, and distribution reduction.  Based on these properties the authors are able to propose an algorithm for efficient computation of the privacy parameters of smoothed differential privacy. Ultimately the authors evaluate experimentally the proposed framework of smoothed differential privacy and conclude that discrete sampling mechanisms are private in real-world elections as well as some artificial neural networks can be private without adding additive noise.

**Audience:**

Yes

**Broader Impact Concerns:**

I am really not familiar with the field, but the authors could potentially devote a small paragraph and discuss broader impacts at the end of the paper.

**Claims And Evidence:**

No

**Requested Changes:**

Please see the weaknesses mentioned above.  I think this paper needs a major revision before it can be accepted.

**Strengths And Weaknesses:**

**Strengths**

The paper proposes a framework that can be seen as more realistic, as it does not use the pessimistic estimates of standard differential privacy.  The paper has both theoretical results as well as an experimental study, thus elucidating the proposed framework in both interesting ways.

**Weaknesses**

In my opinion this paper assumes basic knowledge of differential privacy from the reader and is not an entirely stand-alone paper. I, for one, am not really familiar with differential privacy and I am not confident about my understanding in various cases.  However, let me come to some concrete issues that the paper has, which I believe are issues even if the reader knows differential privacy.

- Differential privacy is defined in page 4 but some intuitive notion of what it actually is would help the reader a lot in the beginning to understand the existing framework (differential privacy), the proposed framework (smoothed differential privacy), and appreciate better the work of the authors.

- Instead the authors discuss in the first sentence methods of how DP can be achieved, as well as have some forward referencing the makes it hard to follow what kind of message the authors are trying to convey. For example, in line 33 the privacy parameter $\delta$ is mentioned, but no intuition given as to what $\delta$ really measures. We only understand as readers that smaller values of $\delta$ are more desirable, but essentially this is it.

- The paragraph to the left of Figure 1 needs rewriting to convey better meaning.

- The caption of Figure 1 is referencing some equation from the appendix as an explanation.  This needs to change somehow and convey meaning in a caption of a figure without asking from a reader that has just started reading a paper to leave everything on the side and move to the appendix in order to understand something from what should be an "explanatory" figure.

- Also, since we are in figure 1, discussing a bit about adversaries utility in the main text would be nice. In my opinion this is another example that a few meaningful paragraphs of what differential privacy really is, are missing from the introduction of the text.

- While the authors give an example in the introduction, I think the example is only half-successful. It would be nice to have an example with actual histograms and discussion on the manipulation that can be done by an adversary and how information can leak from the database. The way the paper is written it is impossible to understand how something like that can be done.

- Also, I appreciate a lot Tables 1 and 2 that the authors have tried to compile and make it clearer for their work to be understandable. However, again, I think because some high-level discussion is missing from the introduction about DP, it is not really possible to appreciate fully what the authors present. In addition, as a person reading about a topic (DP) that I knew almost nothing beforehand, I find the use of lowercase $x$ to represent some database to be odd.

- In lines 117-118 you try to describe databases and records. I think spending two-three lines and giving an actual example of what you mean it would be nice.  Is it is the case that the record is a single number (or category)?  Is it a vector?  If it can be a vector (which would make more sense for a record), then you can argue a bit more about the nature of $\mathcal{X}$. Then, you could also give an example of neighboring databases $x$ and $x'$ and explain how much of a difference is allowed between two different records.

- Line 172: Theorem statement precedes the definition of the framework of the authors.

- I have a hard time following the argument the authors are trying to make in lines 215-216. Is there any real-world case where one does not use the actual data of a dataset, but rather the gradients for, I suppose, some particular training iteration for some custom artificial neural network architecture?

- The discussion in the experiments needs some more text. Do the authors discuss the right-hand side of Figure 2? Along these lines, in both subfigures, the graph corresponding to $\epsilon = 0.5$ is above the yellow line, which, if I understand correctly (from what is mentioned in the introduction), we would like these graphs to be below the yellow line.  Nevertheless, the authors claim that $\delta$ is exponentially small even if it is above that yellow line.  Some clarification would be in order (I think).

- Overall the authors are allowed to use 12 pages for the main text of the paper and in this case they use 10 pages plus a few more lines. The paper can improve a lot by adding discussion as mentioned above, or by adding a few sentences in other parts where the authors ask from the readers to read the appendix (e.g., caption of Figure 1, or in line 115 where they refer the readers for reading part of the related work in the appendix).



**References**

I would prefer to see a reference being cited for the claim in line 65 (comparison of simplex with other polynomial methods).


**Some Typos**

Without referring to a specific line, I think that in most of the cases where the authors use "noises" I would actually use the term "noise".

Line 20: mechansim -> mechanism
Line 49 (second paragraph, line 4): sampling -> Sampling (capitalize)
Line 121: What do you mean by "image space"? Shouldn't this be defined somewhere?
Line 162: opposites to -> contrasts

---

> ### Author Response · Authors · 2023-09-09
> **Authors' Response To Reviewer zTAA**
>
> Thank Reviewer zTAA for the comments and suggestions! We are very willing to further collaborate with the reviewer to make the paper more friendly to general readers (maybe without DP background). As suggested by TMLR, we will provide the revision of this paper after receiving all reviews (See **3. Rebuttal and discussion** of the **Review process** section in https://jmlr.org/tmlr/editorial-policies.html ).
>
>
> **[Presenting DP in the beginning]** We agree that adding a brief introduction about DP would help the presentation. In the revision, we plan to include a brief introduction about DP (with a pointer to our detailed discussion to avoid duplication) between lines 31 and 32.
>
>  *The standard notion of DP measures the worst-case privacy leakage (See Section 2 for the formal definition and detailed discussions). At a high level, DP considers the worst-case input and worst-case output of the (random) mechanisms. $\epsilon$ and $\delta$ are the privacy parameters to measure the privacy leakage in the worst-case described above. At a high level again, $\epsilon$ is a mechanism-designer-decided threshold for the “acceptable amount of privacy leakage”, while $\delta$ measures the probability that the threshold got exceeded. Thus, smaller $\epsilon, \delta$ represents stronger privacy guarantees. The usual requirements upon a private mechanism are $\epsilon = O(1)$ and $\delta = o(1/n)$ (since $\delta$ measures the “failure probability”).*
>
> **[The privacy parameter is mentioned without intuition about what it measures]** The intuition is also included in the paragraph above
>
>
> **[The paragraph to the left of Figure 1 needs rewriting to convey better meaning]** We are not sure what the reviewer expects us to improve / to convey in the paragraph to the left of Figure 1. Any further clarifications will be very appreciated!
>
>
> **[Figure 1 references “adversaries’ utility” in appendix]** Thanks for the suggestion! We will replace the pointer to the appendix with the following sentence: *The adversaries’ utility represents the adversaries’ Bayesian utility in predicting a single voter’s votes in US presidential elections.*
>
>
> **[Discussing a bit about adversaries' utility in the main text would be nice]** Thanks for the suggestion! In the revision, we will move the majority of Appendix B to the main text to introduce adversaries’ utility. The added part will also serve as an additional justification for DP and Smoothed DP.
>
>
> **[It would be nice to have an example with actual histograms and discussions on the manipulation that can be done by an adversary and how information can leak from the database]** The adversary is not manipulating the histogram : ), instead, the adversary infers the votes by observing the output (the election result). We thus believe the explanation should be clear after adding the high-level descriptions about $\delta$. Any further comments from the reviewer will be very appreciated!
>
>
> **[Missing high-level discussion about DP in Table 1 and Table 2]** In addition to the revisions introduced above, we will add a pointer to the formal definition of DP in Table 1 and Table 2. Any further suggestions from the reviewer about the presentation will be very appreciated!
>
>
> **[Lowercase $x$  to represent database looks odd]** We adopt the same notation system as [Dwork et al., 2014] (one of the most famous books about DP). The high-level idea of this notation system is: unlike sets (using \mathcal, like $\mathcal{S}$), databases allow repeated entries.
>
>
> **[Giving an actual example of databases and records would be nice]** We agree that adding an example can make this paper more friendly to readers without database or DP background. The following sentence will be included in the revision. *In the motivating example, $\mathcal{X}$ is a set containing $0$ and $1$ only, where $0$ represents a vote to one candidate while $1$ represents the other one. Database $x$ represents all voters’ votes (a $n$-dimensional binary vector). A record represents a single dimension (e.g., the $i$-th dimension) of $x$. Two databases are considered to be neighboring if no more than one voter’s vote is different.*
>
>
> **[Line 172: Theorem statement precedes the definition of the framework of the authors]** Note that Theorem 1 is about the vanilla differential privacy instead of smoothed DP. DP is defined in Definition 1 and $\delta_{\epsilon,\mathcal{M}}(x)$ is defined in Definition 2. Both of them appear before the theorem statement.  We will make it clearer in the revision.

---

> > ### Comment · Reviewer_zTAA · 2023-09-18
> > **Thank you for the response**
> >
> > I think these are all very useful revisions/clarifications if done and will help to improve the readability of the paper and the readers will be able to appreciate your results better.  In the revised version please use all 12 pages that you have available and provide a better exposition of the background knowledge and of your results.  I am not entirely sure if moving Appendix B (or, its majority) into the main text is the best solution, but there are definitely points that need to be explained so that the caption of Figure 1 is understandable and can stand alone in the main text of the paper without any reference in the appendix (other than further clarification and additional details).
> >
> > I believe that if you do the above, then you will have to restructure the paragraph that I ask for restructuring.  That is the paragraph that is referring to Figure 1 and the text surrounding the first reference to Figure 1 does not provide (at least to someone like me) all the necessary information needed to understand what is actually presented in that figure.

---

### Review · Reviewer_m9fG · 2023-09-10

**Summary Of Contributions:**

The paper proposes a novel extension notion of DP, named smoothed differential privacy, to measure the privacy leakage of mechanisms without adding additional noises that do not satisfy DP. By noting that the worst-case privacy according to DP might be too loose to serve as a practical measure for evaluating and comparing mechanisms with sampling noise (while without additive noise) in real-world applications (e.g. election with sampling noise), the smoothed DP with basic properties (e.g., post-processing and composition properties)  is introduced. Further, an algorithm used to calculate the privacy parameters for smoothed DP is proposed.

**Audience:**

Yes

**Claims And Evidence:**

Yes

**Requested Changes:**

1. Could the author give some explanation or insights when introducing the definition of smoothed DP in Section 3.2? For example, I was confused as to why the set of distribution $\Pi$ was introduced. Does this come from smoothed analysis? The original DP holds for any $x$ and $x’$, while the smoothed DP seems to restrict the distribution of $x$ to a small set?

2. Theorem 3 (i) shows that when $\epsilon \ge \ln(1/1-\eta) +c$,  $M_H$ satisfies a certain level smoothed DP. Does this imply that $M_H$ is nearly non-private even under the notion of smoothed DP if the sampling rate $\eta$ tends to 1 or the constant $c$ is very large?  In addition, could the authors give some explanation or a discussion about the constant $c$?
Also, in line 299, the authors state that SHM is $(3,\exp(-\Theta(n)),\Pi)$-smoothed DP, it seems that $c$ and $m$ are ignored?

Minor comments

1. The notion $\Theta(\cdot)$ and $\omega(\cdot)$ are not introduced.

2. In line 251, “The application” should be “the application”. In line 80, “pre-prpcessing” should be “pre-processing”.

**Strengths And Weaknesses:**

Strengths
1. The paper introduces the notion of smoothed DP following the worst average-case idea behind the smoothed analysis. The proposed notion can be used to measure the privacy leakage of some types of non-noising mechanisms (e.g., mechanisms based on SHM including ML algorithms based on randomly sampling training data and deterministic voting rules), which is very interesting.

2. Some basic properties of smoothed DP (especially pre-processing property) are introduced. Based on these, the authors used smoothed DP to measure the privacy of some commonly used mechanisms, where sampling noise is unavoidable.

3. Experiments support the findings and the results of the paper.

Weakness

1. Smoothed DP can only address some specific problems or scenarios, such as mechanisms with sampling noise. While DP, RDP or GDP are more general.

2. The running time of calculating the exact privacy profile $\delta$ for smoothed DP might be very large (Theorem 2 shows that the running time is $O(n^{2m+\ell*-3}+\ell \log(\ell*)) + Cpl_{DP}$).

Overall, the paper is well-written and the idea is novel and interesting.

---

> ### Author Response · Authors · 2023-09-18
> **Authors' Response to Reviewer m9fG**
>
> Thank Reviewer m9fG for the comments! As suggested by TMLR, we will provide the revision of this paper after receiving all reviews (See **3. Rebuttal and discussion** of the **Review process** section in https://jmlr.org/tmlr/editorial-policies.html ).
>
>
> **[Smoothed DP can only address some specific problems or scenarios, such as mechanisms with sampling noise. While DP, RDP or GDP are more general.]**
> Smoothed DP is a general definition. One can talk about the smoothed DP parameter for every mechanism w.r.t. any smoothing set of distributions $\Pi$.   Our discussion focused on sampling noise because it is an example where the smoothed DP parameter is orders of magnitude smaller than the corresponding DP parameter (see the captions of Table 1 in Section 1).
>
> Also, Smoothed DP relaxes DP in a complementary dimension compared to RDP or GDP (see Table 2).  Smoothed DP relaxes the distribution over the database $x$ (replace worst-case analysis with smoothed analysis), while RDP/GDP relaxes the worst-case output (replace worst-case analysis with average-case analysis).
>
> By the way, we chose to define smoothed DP using privacy profile $\delta$ is a deliberate choice. Smoothed f-DP and smoothed GDP are straightforward extensions of Definition 3.  Renyi DP is not directly compatible since it does not use $\delta$ as its privacy parameter. One could also think about coming up with a smoothed RDP by applying smoothed analysis upon privacy parameters other than $\delta$.
>
>
> **[The running time of calculating the exact privacy profile for smoothed DP might be very large]** As stated in lines 251-253, in most application scenarios of smoothed DP,  calculating the privacy profile $\delta$ of smoothed DP only requires a polynomial extra time than DP.
>
>
> **[I was confused as to why the set of distribution $\Pi$ was introduced. Does this come from smoothed analysis?]** Yes, it comes from the smoothed analysis (see Lines 72 and 176-177). We will further emphasize it in the revision.
>
>
> **[The original DP holds for any $x$ and $x’$, while the smoothed DP seems to restrict the distribution of $x$ to a small set?]**
> Smoothed DP does not restrict the distribution of $x$ to a small set. Instead, it allows the distribution of database $x$ (the vector $(\pi_1,\cdots,\pi_n)$) to have non-zero mass on all possible databases. For example, when the set of distributions $\Pi$ is strictly positive (the assumption of Theorem 3), each entry’s (each $X_i$’s) distribution has non-zero mass on all possible choices. Then, the distribution of database $x$ has non-zero mass on all possible databases.
>
> We are not 100% sure about what the reviewer means for “small set” here. If it means $\Pi$, we would like to note that the set of distributions $\Pi$ can be very large or even an infinity set (see lines 230-234).
>
> Finally, let us note that both DP and smoothed DP consider the worst-case neighboring database $x’$ and the worst-case output $\mathcal{S}$ (see Table 2).
>
>
> **[Does Theorem 3 imply that $\mathcal{M}_H$ is nearly non-private even under the notion of smoothed DP if the sampling rate tends to 1 or the constant $c$ is very large?]**
> 1. When the sampling rate tends to 1, yes,  $\mathcal{M}_H$ is nearly non-private. This justifies the smoothed DP notion because $\mathcal{M}_H$ contains no noise (no randomness) when the sampling rate is 1. In other words, the histogram is directly published, which is expected to be non-private.
> 2.  When constant $c$ is large, no, the $\delta$ parameter of $\mathcal{M}_H$ is private. As shown in Remark 5, a larger $c$ (or a larger $\epsilon$) corresponds to a smaller $\delta$ (still exponentially small).
>
>
> **[Could the authors give some explanation or a discussion about the constant $c$?]** As shown in Line 290, $c$ is the difference between the mechanism-designer-given $\epsilon$ and $\ln(\frac{1}{1-\eta})$. If $c$ is a constant, part (i) of Theorem 3 can be applied (otherwise not). We will provide more intuition about $\epsilon$ in the revision.
>
>
> **[in line 299, the authors state that SHM is $(3, \exp(-\Theta(n)), \Pi)$-smoothed DP, it seems that $c$ and $m$ are ignored?]** This is because the constants are not considered in the big $\Theta$ notation (see https://en.wikipedia.org/wiki/Big_O_notation#Family_of_Bachmann%E2%80%93Landau_notations ). We will further emphasize that both $c$ and $m$ are constants in the revision.
>
>
> **[The notion $\Theta(\cdot)$ and $\omega(\cdot)$ are not introduced]** Both of them are the standard asymptotic notions for complexity (see https://en.wikipedia.org/wiki/Big_O_notation#Family_of_Bachmann%E2%80%93Landau_notations ). We will explicitly define them in the revision.
>
>
> **[Typos]** Thanks for pointing out our typos! All of them will be fixed in the revision.

---

### Review · Reviewer_zUmC · 2023-09-28

**Summary Of Contributions:**

This paper presents a novel concept in differential privacy, termed smooth differential privacy. Unlike differential privacy, smooth differential privacy stipulates that the expected privacy profile must remain bounded across all given data distributions. The authors illuminate a reduction property of smooth differential privacy, indicating that the vertices of the convex hull of data distributions determine its satisfaction. To illustrate the practical application of smooth differential privacy, a sampling-histogram mechanism is presented. This mechanism generates a histogram from a uniformly random subsample of a defined size. It is demonstrated that this mechanism adheres to smooth differential privacy when it meets the f-strictly positive condition. The authors also observe that the privacy profile derived from this is most effective under specific conditions. The study also reveals that subsampling-based mechanisms fall short of achieving standard differential privacy with discrete-valued data and smooth differential privacy with continuous-valued data. Experimental results emphasize the superior privacy profile of the proposed mechanism, particularly when tested with elections and 1-step quantized SGD.

**Audience:**

Yes

**Broader Impact Concerns:**

I have not found any concerns regarding border impact.

**Claims And Evidence:**

No

**Requested Changes:**

(Major) Please elucidate the threat model, inclusive of adversaries' capabilities, under which smooth differential privacy can assure privacy.

(Major) Should the authors show the adversary's utility depicted in Figure 1 as reasonable, it is requisite to elucidate its reasonableness within the main text.

(Major) It is essential to articulate the necessity of employing smooth differential privacy. To appraise the privacy protection that subsampling furnishes, the original differential privacy seems sufficient.

(Major) It is imperative to significantly enhance the manuscript by rectifying linguistic inaccuracies.

(Major) Although the authors assert that pre-processing is a distinctive characteristic of smooth differential privacy, a similar characteristic may hold in original differential privacy as well.

(Minor) Table 2 presents an unfair comparison. The term "smoothed analysis" should be replaced with "less informative adversary."

(Minor) It is necessary to elucidate why the selections of π illustrated in the experiments render reasonable privacy protection.

(Minor) I find Justifications 1 and 2 to be untenable as $\delta(x)$ can take a large value under smooth differential privacy.

(Minor) The manuscript could be further enhanced by introducing a discussion on integrating the subsampling-based mechanism with anonymization algorithms, including those aimed at k-anonymity. An anonymized dataset may invariably fulfill f-strict positivity.

**Strengths And Weaknesses:**

### Strengths

The examination of privacy guarantees against a realistic adversary is well-motivated and is believed to hold interest for readers of TMLR. Differentially private mechanisms are designed to protect the privacy of a dataset against adversaries who, hypothetically, have knowledge of the entire dataset except for a single bit of information. While these defenses are robust against side-channel attacks, it is conservative to assume that such an all-knowing adversary exists in real-world situations. Investigating a realistic adversary and devising defenses to counter such adversaries is a promising avenue of exploration.

Exploring the privacy implications of subsampling is captivating. Subsampling can potentially conceal specific types of information from the original dataset. Understanding what subsampling can and cannot conceal may aid in advancing the development of privacy mechanisms. In this context, the authors effectively present some extent of privacy guarantee through the sampling-histogram mechanism.

Beyond ensuring smooth differential privacy, the authors have validated the tightness of the privacy parameter ensured by the sampling-histogram mechanism. Even though this tightness is demonstrated specifically when both $m$ and $f(m,n)$ are of a constant order, such parameter settings are plausible in realistic scenarios. Through this tightness analysis, the authors have successfully delineated the privacy guarantees offered by the sampling-histogram mechanism.

### Weaknesses

One of the primary shortcomings of the manuscript is its unsubstantiated claim regarding realistic models. Although the authors contend that they have elucidated a method for measuring privacy within "realistic models," they provide no definitive evidence to support the assertion that the proposed smooth differential privacy is secure against realistic adversaries. Notably, smooth differential privacy provides security only against adversaries of lesser potency than does the original differential privacy. This is because smooth differential privacy does not intrinsically guarantee differential privacy. While differential privacy assures security against the strongest adversaries, smooth differential privacy lacks this assurance. This discrepancy leads to concerns that some realistic adversaries might circumvent the protections offered by smooth differential privacy. Hence, providing the threat model, including adversaries' abilities, under which smooth differential privacy can guarantee privacy is crucial. Such clarity is vital for verifying the claims of privacy protection in realistic models.

To illustrate their point, the authors present in Figure 1 a comparison between the privacy profiles of original differential privacy and smooth differential privacy in the context of US presidential elections. They suggest that the proposed privacy measure correlates exponentially with the adversary's utility. However, the adversary's utility appears to have been selectively chosen by the authors to align with the privacy profile of smooth differential privacy. The correlation shown in Figure 1 does not inherently validate the practicality of smooth differential privacy.

The sampling histogram mechanism might satisfy traditional differential privacy, provided that all datasets conform to $f$-strict positivity. To encapsulate the privacy assurances offered by subsampling, the original differential privacy appears adequate. Therefore, the introduction of a novel—yet unsubstantiated—privacy concept, namely smooth differential privacy, is questionable.

Smooth differential privacy agrees with standard differential privacy when $\Pi$ is the set of all distributions, as this set includes one supported on a specific dataset. It is self-evident that the privacy profile of smooth differential privacy is equivalent to or lesser than that of differential privacy. Therefore, the rationale behind comparing smooth differential privacy with standard differential privacy, as presented in Section 6, is unclear to me.

Moreover, the manuscript is permeated with linguistic inaccuracies, significantly detracting from its overall quality.

---

> ### Comment · Reviewer_zUmC · 2023-11-08
> **Appreciation for the author's response**
>
> I appreciate the authors' response. The revised manuscript has significantly improved in clarity regarding security from the initial submission. However, I remain unconvinced that the main claim is sufficiently supported. According to the revised manuscript, the difference in the threat model between smoothed and conventional differential privacy lies in the adversary's ability to select the dataset, a point with which I concur. This introduces a concern that the chosen dataset may be unrealistic, leading to a contradiction with the claim of a privacy guarantee in realistic situations. To ensure the accuracy of the manuscript, it is essential to either present ample evidence supporting the feasibility of the dataset selection or to refine the claims accordingly.
>
> The example provided is unfair to conventional differential privacy. Although f-strict positivity is assumed to comply with smooth differential privacy, the dataset presented in the example does not meet this criterion. The authors stated in their response to my review that even if the dataset satisfies f-strict positivity, no mechanism can guarantee conventional differential privacy. Thus, the current example is less effective in demonstrating the limitations of conventional differential privacy.

---

> > ### Author Response · Authors · 2023-11-08
> > **Further Clarifications**
> >
> > We thank the reviewer for the reply! However, we believe the second paragraph of the reviewer's reply still includes several misunderstandings of this paper.
> >
> > **[We did *NOT* claim "no mechanism can guarantee conventional differential privacy"]** Instead, we claimed that the sampling-histogram mechanism is smoothed differentially private but not differentially private (under certain conditions, See Table 1, Theorem 3, and Lines 2, 10, 32-38, 43, just list a few). We would like to point out that "no mechanism can guarantee conventional differential privacy" is an incorrect claim. As analyzed in Table 1, if additive noise is added, the mechanism will be private under both DP and smoothed DP.
> >
> >
> > **["Dataset satisfies f-strict positivity" is not a well-defined statement]** As mentioned in our response, $f$-strict positivity is **NOT** a property of the database but for the set of distributions $\Pi$ (see Lines 283-287). As stated in Clarification 1 in our previous response: the mechanism should be private if the worst-case database considered in DP happens with extremely low probability, where the probability is determined by $\vec\pi$ (selected according to $\Pi$).  $f$-strict positivity does **NOT** rule out the worst-case database. Instead, it guarantees the worst-case database to appear with extremely low probability.
> >
> >
> > **[$f$-strict positivity is not an assumption for the smoothed DP's definition]** Instead, $f$-strict positivity is just for Theorem 3(i). As stated in Definition 3, smoothed DP does not require any assumptions on $\Pi$. Also note that Theorem 3(i) does **NOT** mean smoothed DP cannot be achieved if $\Pi$ is not strictly positive. The main message of Theorem 3(i) is: it guarantees the smoothed DP property of the sampling-histogram mechanism when $f$-strict positivity. More examples can be found in Table 1.

---

### Author Response · Authors · 2023-12-12
**The camera-ready version has been submitted**

We thank the Action Editor and all reviewers for the helpful comments!

The camera-ready version of this paper has been submitted.

---

### Decision · Action_Editor_7eZd · 2023-12-03

**Recommendation:** Accept as is

**Comment:**

The paper introduces the concept of smoothed DP, a method designed to measure privacy leakage in mechanisms without additive noises under practical conditions effectively. The authors rigorously establish and demonstrate various properties of smoothed DP. Additionally, they present an efficient algorithm for computing the privacy parameters associated with smoothed DP and validate it through experiments.

The authors have diligently incorporated the reviewers' feedback, resulting in a revised manuscript. Two reviewers endorse its acceptance.  Although one reviewer expresses lingering concerns about the paper's claims, they acknowledge the paper's motivation in addressing privacy guarantees against realistic adversaries, which will engage readers of TMLR. The theoretical analysis presented is also appreciated.

Considering the comprehensive responses from the authors and the reviewers' recommendations, I recommend accepting the paper.

**Audience:**

The subject revolves around differentially private machine learning, a focal point in establishing trustworthy machine learning practices.  It will captivate a broad audience in TMLR.

**Claims And Evidence:**

Yes